# Typhoid intestinal perforation in Francophone Africa, a scoping review

**Leah Sukri** [1]*, **Audry Banza**[2], **Katherine Shafer**[2], **Yakoubou Sanoussi**[2], **Kathleen M. Neuzil**[1], **Rachid Sani**[3]

1 Center for Vaccine Development and Global Health, University of Maryland School of Medicine, Baltimore, Maryland, United States of America, 2 Département de Chirurgie, Hôpital de la SIM, Galmi, Niger, 3 Département de Chirurgie et Spécialités Chirurgicales, Hôpital National de Niamey, Niamey, Niger

* Leah.sukri@som.umaryland.edu

**Data Availability Statement:** All data used for this study is included as supporting information file labeled "S1 Table".

**Funding:** This publication is based on research funded in part by the Bill & Melinda Gates

## Abstract

Typhoid intestinal perforation (TIP) is a leading cause of peritonitis and indication for emergency surgery in Africa, with reported mortality rates up to 30% in pediatric patients. Currently, data on TIP in Western databases are primarily from countries that speak English, likely due to non-English publication and citation biases. Despite the high burden of infectious diseases in Francophone Africa, data from these countries regarding TIP remain limited. This study aims to highlight the incidence and morbidity of TIP in Francophone African countries using an extended search algorithm. We conducted a scoping review using the PubMed, EMBASE, and SCOPUS databases with the keywords "peritonitis", "non-traumatic ileal perforation", and "typhoid" in Francophone African countries. Additionally, we contacted surgeons in Africa and concurrently used citation chasing to obtain data not found in western databases. In total, 32 studies from 12 countries were identified and included in this review. A total of 22 publications were in French. Patient median age was 20 years and TIP caused a median of 35% of acute peritonitis cases. Mortality rates ranged from 6–37% (median: 16%). Rate of complications ranged from 15–92% (median: 46%). Ileostomy creation as a treatment for TIP varied between hospitals (0–79%), with the highest rates reported in Niger. In Francophone Africa, TIP is associated with high morbidity and mortality, most commonly in children and young adults. Interventions, including improved sanitation and the introduction of typhoid conjugate vaccines into routine vaccination programs, have the potential to significantly decrease typhoid fever and its complications.

## Introduction

Typhoid fever is caused by *Salmonella enterica* serovar Typhi (*S.* Typhi), a gram-negative bacterium primarily transmitted through the consumption of contaminated water or food [1]. In 2019, there were more than 9 million cases of typhoid fever and approximately 110,000 deaths worldwide [2]. Children are disproportionately impacted, with 53.6% of all typhoid deaths occurring in patients under 15 years old [3]. Furthermore, the prevalence of typhoid fever is highest in low-and-middle-income countries that lack safe water, sanitation, and hygiene (WASH) infrastructure [3].

Foundation (Investment ID INV-030857 to KMN). The funders had no role in study design, data collection and analysis, decision to publish, or preparation of the manuscript. The findings and conclusions contained within are those of the authors and do not necessarily reflect positions or policies of the Bill & Melinda Gates Foundation.

**Competing interests:** The authors have declared that no competing interests exist.

Typhoid intestinal perforation (TIP) is a late complication of typhoid fever that generally occurs approximately two weeks after initial symptoms [1]. Patients present with fever and non-specific symptoms, such as abdominal pain, rebound tenderness, and vomiting [4, 5]. One or more oval anti-mesenteric perforations in the distal ileum, as described in operative reports, is considered pathognomonic for TIP [6, 7]. While rare in developed countries, TIP remains a common surgical condition in typhoid-endemic settings, particularly amongst children [7–9].

Prior to 2018, uptake of typhoid vaccines was low and typically limited to travelers and outbreak control. This changed in 2018, when a single dose of typhoid conjugate vaccine (TCV) was recommended by the World Health Organization (WHO) for routine use in children 6 months of age and older in endemic countries [1]. Three large clinical trials of TCV were conducted amongst children aged 9 months through 12–15 years in Bangladesh [10], Malawi [11], and Nepal [12]. A single dose of TCV was demonstrated to be safe with an efficacy rate of 79–85% across the studies. Additionally, TCV remains efficacious over four years [13]. TCV can be safely co-administered with routine childhood immunizations to include measles-rubella, polio, meningococcal capsular group A conjugate, yellow fever, and human papillomavirus vaccines [14]. Yet, limited data on country-specific typhoid burden and its complications remains a major barrier to vaccine introduction in many regions.

There are 21 Francophone African countries, representing 25% of the total population in Africa [15]. Socioeconomic, infrastructure, and language barriers present additional challenges to most of these countries. Insufficient WASH infrastructure and taxed health systems lead to a higher burden of infectious diseases in these countries, compared to non-Francophone countries [16]. Definitive data on disease burden in many Francophone countries is lacking, due in part to limited funding for healthcare resources and to the availability of information in French [17].

Understanding the relationship between typhoid fever and TIP has immediate implications for decision-making and public health policy regarding TCV introduction. In recent reviews on TIP in Africa, Francophone countries were underrepresented [7, 18]. To address this knowledge gap, we conducted a scoping review on the incidence of TIP in Francophone Africa and examined the rate of TIP in the pediatric population given the WHO recommendations for use of TCV in children and its potential to reduce the incidence of this life-threatening surgical disease.

## Materials and methods

This scoping review followed the Arksey and O'Malley framework for scoping studies [19]. A search strategy was developed in consultation with surgeons from Francophone African countries, experts in typhoid fever, and University of Maryland, Baltimore librarians. We searched EMBASE, PubMed, and SCOPUS databases using keywords such as "peritonitis", "non-traumatic ileal perforation", and "typhoid" along with individual Francophone country names that included Benin, Burkina Faso, Burundi, Cameroon, the Central African Republic, Chad, Comoros, Congo, Côte d'Ivoire, the Democratic Republic of the Congo, Djibouti, Equatorial Guinea, Gabon, Guinea, Madagascar, Mali, Niger, Rwanda, Senegal, the Seychelles, and Togo.

We concurrently contacted African surgeons and other local health leaders and conducted citation chasing. We included all full text studies in Francophone Africa, with a reported rate of TIP, published from 1 January 2000 through 12 January 2023. Studies were excluded if data sets were duplicates or the age range of participants was not reported. French language articles were translated into English before data extraction.

Data, including author names, date of publication, study location, study period, number of patients, age range, mortality rate, morbidity rate, and ileostomy creation rate, were extracted

and entered into a Microsoft Excel sheet. If specific data were not reported in the study, the variable was left blank. We summarized findings using Microsoft Excel.

## Results

A total of 6052 articles were identified with 4736 titles and abstracts screened after the removal of duplicates (Fig 1). After screening, 215 manuscripts were assessed for eligibility with 39 articles that met the inclusion criteria; 7 were subsequently excluded, as 4 did not report participant age and 3 were sub-studies already included in the review. In total, 32 manuscripts were included: 10 in English, 22 in French (S1 Table). Nineteen manuscripts were identified through the database searches and 13 through citation chasing and local outreach.

Twelve countries are represented in this review with studies conducted in 28 hospitals. Benin, Burkina Faso, and Niger were the most represented countries with 5, 5, and 6 studies, respectively. The rest of the countries had three or fewer studies. Study years ranged from 1990–2019, with 79% of studies conducted after 2000.

Table 1 summarizes the number of patients, age, incidence, and complication rate of TIP per country for all age groups. Data from individual studies can be found in S1 Table. The median age was 20 years. The number of TIP patients ranged from 7–2931 per study and TIP accounted for a median of 35% of acute peritonitis (range 15–92%). The median mortality rate was 16% (range 6–37%) while median morbidity rates was 46% (range 15–92%). Ileostomy as a treatment for TIP ranged from 0–79%. There were no trends based on study year.

A total of 15 articles from 10 countries reported on pediatric and young adult populations ranging from 0 to 20 years (Table 2). Among this population, TIP accounted for a median of

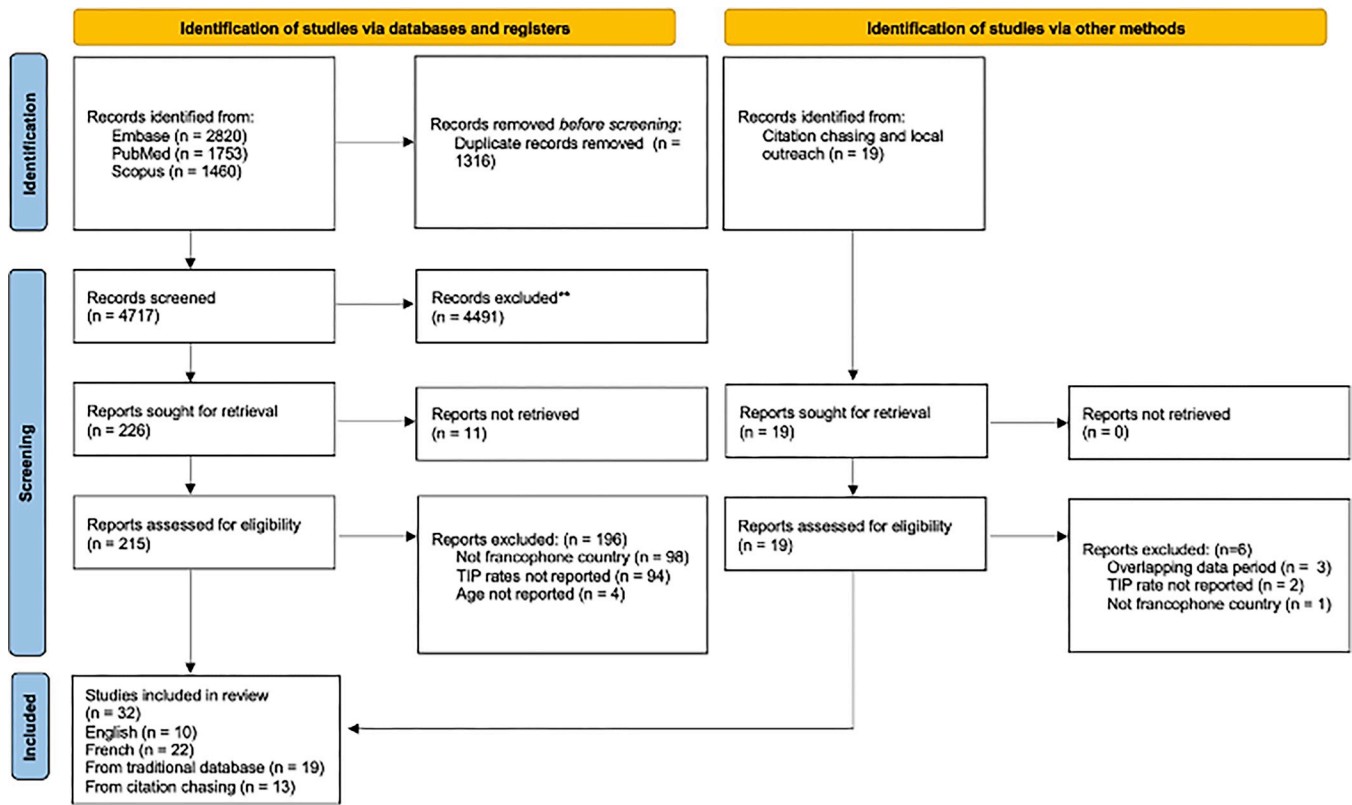

**Fig 1. PRISMA chart of studies identified, screened, and included.**

**Table 1. Characteristics of TIP for all included studies grouped by country.**

| Country | Number of included studies | Average age in years (range in studies) | Number of TIP (% peritonitis) | % Postop morbidity | % Postop mortality | % Ileostomy creation |
|---|---|---|---|---|---|---|
| Benin | 5 [20–24] | 12–23 (2–75) | 28–104 (35–64%) | 15–49% | 7–31% | 0–14% |
| Burkina Faso | 5 [25–29] | 14–30 (1–77) | 29–216 (20–43%) | 35–72% | 10–17% | 15–79% |
| Cameroon | 3 [30–32] | 24–38 (3–70) | 17–43 (14%) | 58% | 16–37% | 11% |
| Central African Republic | 1 [33] | (10m – 7) | 31 | 16% | 29% | 0 |
| Chad | 1 [34] | 26 (15–70) | 72 (15%) | | 11% | 46% |
| Cote d'Ivoire | 3 [35–37] | 9–34 (3–69) | 48–82 | 46–92% | 6–34% | 6–52% |
| Democratic Republic of the Congo | 1 [38] | 19.5 | 39 (35%) | 80% | 13% | 13% |
| Guinea | 1 [39] | 20 (4–41) | 7 | | 14% | 0 |
| Mali | 3 [40–42] | 10–23 (2–65) | 105–385 (32%) | 22–37% | 8–16% | 8–16% |
| Niger | 6 [43–48] | 10–23 (0–95) | 56–2931 (35–69%) | 31–46% | 11–29% | 4–68% |
| Rwanda | 1 [49] | (1m – 15) | 11 (17%) | 91% | 27% | |
| Togo | 2 [50, 51] | 10–20 (3–75) | 110–112 (68%) | 45% | 16–21% | 13% |

56% of acute peritonitis with post-operative mortality rates of 4–33% and morbidity rates of 16–91%. Only six studies reported ileostomy rates in this population, which ranged from 0–63%.

## Discussion

This scoping review describes the prevalence of TIP in Francophone Africa. Although the number of TIP cases varied across countries, typhoid accounted for a large proportion of acute peritonitis in this region and was a major cause of emergency abdominal surgery. Consistent with prior studies, TIP remains a disease of the pediatric and young adult populations [7, 18]. In 75% of the included articles, the average age amongst TIP patients was under 25 years. The median case fatality rate (CFR) amongst Francophone countries is 16%, which is comparable to a historical CFR of 20% for TIP cases in Africa [7]. In pediatric patients, the TIP CFR in Francophone Africa was 4–33%, which remains consistent with the previously reported CFR of 11–30% amongst children in Sub-Saharan Africa [18].

Morbidity of TIP cases was extremely high, likely due to resource limitations in Francophone African countries where access to appropriate and timely healthcare is often difficult. As a result, patients present with severe disease, which is associated with an increased number of post-operative complications. The most common complications reported in prior studies include surgical site infection, wound dehiscence, and enterocutaneous fistula, all of which increase length of hospital stay and place a significant burden on families [7, 52]. Furthermore, increased disease severity is associated with higher degrees of peritoneal contamination. In severe cases, surgeons often perform damage control surgery, requiring patients to return to the operating room several times during their hospital stay. Additionally, simple suture repair or bowel resection is oftentimes not appropriate in these severe cases due to high risk of further

**Table 2. Characteristics of TIP for pediatric patients in included studies.**

| Country | Author | Study Period | Age in years (average) | Number of TIP | % TIP pediatric patients | % Postop complications | % Postop mortality | % Ileostomy creation |
|---|---|---|---|---|---|---|---|---|
| Benin | Lo Sorto et al [20] | 2002–2003 | 4 to 15 | 24 | 55% | 10 (42%) | 8 (33%) | 0 |
| | Sambo et al [21] | 2015–2016 | 4 to 15 | 22 | 79% | | | |
| | Tobome et al [24] | 2018–2019 | 2 to 19 | 16 | 40% | | | |
| Burkina Faso | Ouedraogo et al [25] | 2010–2014 | 2 to 19 | 122 | 56% | | | |
| Cameroon | Johnson Alebeleye et al [31] | 2017–2018 | 12 to 19 | 9 | 24% | | | |
| Central African Republic | Bobossi Serengbe et al [33] | 1997–1998 | 10m – 15 | 31 | | 5 (16%) | 9 (29%) | 0 |
| Cote d'Ivoire | Kouame, et al [36] | 1990–2000 | 3–16 (9) | 48 | | 22 (46%) | 3 (6%) | 3 (6%) |
| Guinea | Mallick & Klein [39] | 1993–1998 | 4–8 | 2 | 29% | | | 0 |
| Mali | Togo et al [40] | 1999–2008 | 2 to 15 | 250 | 65% | | | |
| | Coulibaly et al [42] | 2005–2010 | 3–14 (10) | 105 | | 39 (37%) | 16 (15%) | 31 (30%) |
| Niger | Harouna et al [43] | 1995–1996 | 4 to 20 | 26 | 46% | | 3 (12%) | |
| | Adamou et al [46] | 2013–2015 | 0–15 (10) | 153 | | | 22 (14%) | 96 (63%) |
| | Adamou et al [44] | 2013–2019 | 1 to 15 | 2113 | 72% | | | |
| Rwanda | Mutabazi et al [49] | 2015–2016 | 1m – 15 | 11 | | 10 (91%) | 3 (27%) | |
| Togo | Kassegne et al [51] | 2009–2011 | < 15 | 61 | 56% | | | |

infection, necessitating the creation of ileostomies [53]. Although inconsistently reported in previous studies, our review found that ileostomies were frequently used to treat TIP cases, with ileostomy creation rates up to 79% in one study. Ileostomies are difficult to manage and in low-resource settings, complications–such as skin irritation, malnutrition, prolapse, and hernia–are common [53].

Typhoid fever has significant health implications and a high economic burden on patients and their families. In Malawi, the mean cost of inpatient healthcare for typhoid fever was approximately $296 USD where the mean monthly household income is $23.71 USD per person [52]. Costs for TIP patients are much higher and vary widely depending on surgical needs. Ileostomy creation remains the most expensive procedure, with an average cost of $519 USD in one Nigerien hospital, approximately 30% more than comparable TIP surgeries without ileostomies [44]. Ileostomies lead to significant social strain on patients as well. Due to social stigma, women with ileostomies are at a risk of being abandoned by their husbands, leaving them and their children without a source of income. Children with ileostomies often miss more than 1 year of school after their surgery and are at a higher risk of leaving the school system earlier compared to their peers [54]. In resource-limited settings, the introduction of TCV is highly cost-effective [55]. Fortunately, Gavi, the Vaccine Alliance, an international organization that supports equitable and sustainable use of vaccines in the poorest of countries, has

included TCV in their portfolio beginning in 2018. Gavi will fully finance the operational and vaccine costs of a one-time TCV catch-up campaign in children up to 15 years of age in low resource countries, and provide co-financing support including vaccine, vaccination supplies, and operational costs for nationwide introduction into routine immunization programs [56].

To date, six countries worldwide, including three in Africa–Liberia, Malawi, and Zimbabwe–have introduced TCV into their national routine immunization programs [56]. Despite demonstrated safety and efficacy of TCV, the decision to introduce a new vaccine on a national scale is a complicated process. Liberia was the first African country to introduce TCV in April 2021 based on high modeled estimates of typhoid burden [56]. In May 2021, Zimbabwe introduced TCV in response to multiple large typhoid outbreaks and the presence of drug-resistant strains [56]. Malawi's decision to introduce TCV in 2023 was based on a high burden of disease, increasing drug resistance, and known vaccine efficacy [11, 13, 57]. To date, no Francophone African countries have introduced TCV, largely due to competing health priorities and a lack of sufficient country-specific data on the burden of typhoid and its complications, and a lack of knowledge of antimicrobial resistance patterns.

Due to the lack of blood culture availability in many endemic countries, cases of typhoid fever often go undetected, leading to underreported case numbers. As a result, alternative measures for typhoid disease burden are imperative, with TIP and non-traumatic intestinal perforations serving as indicators of disease. In Malawi, the seasonality of intestinal perforations mirrored the seasonality of blood culture-confirmed typhoid [9, 58]. Similarly, monthly intestinal perforations were significantly correlated with monthly positive *S. typhi* cultures in a large typhoid surveillance study in six African countries [5]. In the DRC, surgeons also reported an increase in peritonitis cases during a severe outbreak of typhoid fever, putting additional strain on an already overburdened health care system [59], which could be mitigated with TCV introduction.

Data from Francophone African countries are limited and can be difficult to find due to language biases. Writing for English medical journals may be difficult for many non-English speaking authors. As a result, many will choose to publish manuscripts in their native language [60]. Unfortunately, there is consistent evidence that systematic reviews often exclude non-English manuscripts or fail to identify these papers in the searching and screening phases, neglecting potentially large amounts of viable data and information [17]. Similarly, non-English papers are cited at much lower rates than their English counterparts [60]. This review attempts to address these gaps in information at a country level.

This review has notable limitations. Due to a lack of readily available diagnostics for typhoid fever, there is an inconsistent definition of TIP amongst studies. Only 3 studies diagnosed patients with blood culture confirmation, and most studies used clinical and operative findings to diagnose TIP. As a result, there is variability in study populations. Another limitation is the mix of adult and pediatric populations included in the studies. Previous literature suggests there may be differences in TIP prevalence and progression between children and adults [7, 18]. Finally, this review relied on citation chasing and networking with local surgeons in Francophone African countries to identify TIP manuscripts not available in traditional databases. Thus, despite our broad approach to identifying manuscripts, we likely missed relevant articles in lesser accessed journals. In the future, identifying TIP experts in each country may ensure that each country is appropriately represented. Another possible progression of this study is the creation of a centralized African typhoid database with standardized diagnostic criterion of TIP. This would facilitate further collaboration between Francophone and non-Francophone countries and could also be used to identify or predict areas of typhoid outbreaks and drug resistance throughout the continent.

In this review, we found a high burden of TIP in Francophone Africa. TIP remains a surgical emergency with high morbidity and mortality, resulting in an immense strain on the fragile health care system in these countries. The presence of TIP in a region should serve to catalyze efforts to prevent and treat typhoid fever. While WASH improvements are necessary, these are often difficult and costly to implement and maintain, especially in resource-limited countries. TCVs are proven effective and safe, and vaccination campaigns and routine immunizations remain an advantageous tool, to reach all children, even those in rural and remote areas. TIP may provide the key to bringing TCVs to the most underserved and vulnerable regions, to significantly decrease this preventable disease in children across all of Africa.

## Supporting information

**S1 Checklist. PRISMA checklist.**
(DOCX)

**S1 Table. Data from all included studies included in the review.**
(DOCX)

## Acknowledgments

We would like to acknowledge the University of Maryland Health Sciences and Human Services Library staff for their assistance with development of our search strategy.

## Author Contributions

**Conceptualization:** Kathleen M. Neuzil.

**Investigation:** Leah Sukri, Audry Banza, Katherine Shafer, Yakoubou Sanoussi, Kathleen M. Neuzil, Rachid Sani.

**Methodology:** Leah Sukri.

**Writing – original draft:** Leah Sukri.

**Writing – review & editing:** Leah Sukri, Audry Banza, Katherine Shafer, Yakoubou Sanoussi, Kathleen M. Neuzil, Rachid Sani.

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
