## [Decision Letter · Decision Letter 0]

28 Nov 2023

PGPH-D-23-01879

Typhoid Intestinal Perforation in Francophone Africa, a Scoping Review

Dear Dr. Sukri,

Thank you for submitting your manuscript to PLOS Global Public Health. After careful consideration, we feel that it has merit but does not fully meet PLOS Global Public Health’s publication criteria as it currently stands. Therefore, we invite you to submit a revised version of the manuscript that addresses the points raised during the review process.

We look forward to receiving your revised manuscript.

Kind regards,

Rashi Jhunjhunwala

Academic Editor

Journal Requirements:

Additional Editor Comments (if provided):

Dear Authors,

Congratulations on conducting a thorough scoping review and presenting this important data. Please see comments from the reviewers below.

Reviewers' comments:

Reviewer's Responses to Questions

**Comments to the Author**

1. Does this manuscript meet PLOS Global Public Health’s publication criteria? Is the manuscript technically sound, and do the data support the conclusions? The manuscript must describe methodologically and ethically rigorous research with conclusions that are appropriately drawn based on the data presented.

Reviewer #1: Yes

Reviewer #2: No

2. Has the statistical analysis been performed appropriately and rigorously?

Reviewer #1: I don't know

Reviewer #2: N/A

3. Have the authors made all data underlying the findings in their manuscript fully available (please refer to the Data Availability Statement at the start of the manuscript PDF file)?

Reviewer #1: Yes

Reviewer #2: Yes

4. Is the manuscript presented in an intelligible fashion and written in standard English?

Reviewer #1: Yes

Reviewer #2: Yes

5. Review Comments to the Author

Reviewer #1: PGPH-D-23-01879

Typhoid Intestinal Perforation in Francophone Africa, a Scoping Review

This is a very relevant and important work with high relevance to public health given that typhoid disease burden continues to remain high in LMICs. Countries like Malawi have set a wonderful example by introducing TCV into their national immunization programme and this valuable work by the authors can serve as a pivotal catalyst for other African countries, especially the Francophone settings, in introducing the vaccine as well. Overall, a well written manuscript. Please find some minor comments to consider.

Abstract

• Lines 35-36: Can this be rephrased for more clarity? – Consider: Currently, data on TIP in Western databases are primarily from countries that speak English, likely due to non-English publication and citation biases from non-English settings.

• Line 36: Removed ‘presumed’

• Lines: 39-10: “..to investigate peritonitis, non-traumatic ileal perforation, and typhoid fever in Francophone African countries” – this is unclear. Aren’t these your MeSH search terms. Please rephrase.

• Line 42: Remove ‘traditional’

Introduction

• Line 66: reads well as ‘surgical condition’

• Lines 86-87: Decision making and public health policy related to? I understand this is related to TCV introduction – please specify.

• It would be good to make a mention of how many countries are Francophone in Africa in the Intro – what % of the population in Africa is Francophone?

• Line 89: Isn’t it ‘incidence’ of TIP?

• Line 90-91: “..for use of TCV in children and its potential to reduce the incidence of this life-threatening surgical condition.”

Materials and methods:

• Line 97: Replace ‘a’ with ‘the’.

• Line 98: Remove ‘broad’

• Line 104: Local health leaders – can you be more specific?

Results:

• Figure 1: The last box showing ‘studies included in review’ = 32. Citation chasing was separate 13 articles – if that was the case it does not sum up to 32. Kindly clarify and make it clear in the flowchart – it is confusing for the reader.

• Supplementary table 1: replace ‘study years’ with ‘study period’; if % for TIP was not available, indicate the same a footnote for the table,

• In Figure 2, it should read as ‘country in search with no relevant data available’? Kindly check and clarify.

• Line 134: Remove ‘old’

• This is perhaps beyond the purview of this scoping review—but were you able to look into if there was documentation of a preceding culture confirmed typhoid in these studies with cases of TIP? I think this is important to be mentioned either in methods or discussion, given that blood culture remains the gold standard for diagnosis and a lot many times false positives are common with other tests used in the Dx of typhoid fever, especially in low-income settings.

Discussion:

• In first para of discussion, would you like to add a sentence on specific typhoid disease burden in this region, as this will add to the case of TIP burden in this region? Also, do these regions have access to TCV within their national immunization programme? These are important to set the background for the burden of TIP and its implications which have been well described in the further paragraphs.

• Line 197: Expand Gavi as it has not been expanded before.

• Line 202: Six countries where?

• Line 231: I just saw that that the blood culture aspect that I mentioned above has been addressed here – great! But would be good if you can say if you looked into this aspect during your review and how many such studies reported the same – perhaps challenging but worth mentioning.

Reviewer #2: Excellent scoping review on a critical topic of preventable morbidity and mortality: typhoid intestinal perforation (TIP). The authors provide a scoping review of literature with data on TIP in Francophone African countries, and make a specific effort to highlight articles that are not in English. Their results confirm the need for this type of effort -- 22 of 32 manuscripts are in French, and typically left out of the english-speaking academic arena. These data provide meaningful richness to understanding the massive burden and downstream impact of disease - from damage control surgery to ileostomies to social stigma to catastrophic financial expenditure - that is largely preventable with adequate vaccination programs. The authors' discussion is particularly strong, weaves a coherent narrative and serves as potent advocacy for uptake of vaccination programs.

The methods the authors use a well known and accepted set of guidelines for a scoping review. They additionally performed citation chasing, which is important in these types of reviews and contributed/enhanced their yield significantly.

Results: Their figures and tables are clean and clear.

Discussion: As above, excellent contextualization of study findings within the broader African literature on this topic, the disease management challenges in low-resource environments, policy and vaccination landscapes. Limitations are sincere and well presented.

6. PLOS authors have the option to publish the peer review history of their article (what does this mean?). If published, this will include your full peer review and any attached files.

**Do you want your identity to be public for this peer review?** For information about this choice, including consent withdrawal, please see our Privacy Policy.

Reviewer #1: No

Reviewer #2: No

---

## [Editor Report · Decision Letter 1]

6 Mar 2024

Typhoid Intestinal Perforation in Francophone Africa, a Scoping Review

PGPH-D-23-01879R1

Dear Dr. Sukri,

We are pleased to inform you that your manuscript 'Typhoid Intestinal Perforation in Francophone Africa, a Scoping Review' has been provisionally accepted for publication in PLOS Global Public Health.

Best regards,

Rashi Jhunjhunwala

Academic Editor

Thank you for addressing the comments left by reviewers. We believe this manuscript is ready for acceptance. Congratulations on a well-done scoping review.